# Digital Twin Assistant Active Design and Optimization of Steel Mega-Sub Controlled Structural System under Severe Earthquake Waves

**DOI:** 10.3390/ma15186382

**Published:** 2022-09-14

**Authors:** Zheng Wei, Xun-An Zhang, Feng Sun, William Yi Wang

**Affiliations:** 1School of Mechanics, Civil Engineering and Architecture, Northwestern Polytechnical University, Xi’an 710072, China; 2State Key Laboratory of Solidification Processing, Northwestern Polytechnical University, Xi’an 710072, China; 3Chongqing Innovation Center of Northwestern Polytechnical University, Chongqing 401135, China

**Keywords:** mega-sub controlled structure system, digital twin, finite element model, seismic response

## Abstract

In order to support the best optimized design or strategy based on life-cycle data, the interrelation mechanisms between structure–form and structure–performance should be considered simultaneously and comprehensively besides of the material–property relationship. Here, the structure–property–performance relationship of a designed steel mega-sub controlled structural system (MSCSS) under the reported earthquake waves has been investigated through integrating the finite element simulations and the experimental validations. It can be found that the MSCSS configurations are capable of effectively optimizing the vibration responses with significantly decreased acceleration, which is also much better than the traditional megaframe structure with extra weight. Moreover, if the horizontal connections between the sub- and the megastructures are broken, the displacement of the megastructure will be smaller than that of the substructure. This is because only the vertical connections between the sub- and megastructures work, the larger displacements or the obvious response of the substructures should be caused by the extra weight of the damper on the top floor. It is worth mentioning that the formation of abrupt amplified β of the top floors should be attributed to the sheath effect. Furthermore, the displacement of the substructure is one kind of energy dissipation. Its larger displacement will result in a greater amount of energy dissipation and better performance of the designed configuration. This work supports a digital twin assistant active design and optimization strategy to further improve the control effectiveness of the system and to enhance the mechanical performance of the optimized configuration of MSCSS.

## 1. Introduction

In the data-driven integrated computational materials engineering (ICME) era [1,2,3,4,5], the rapid development of information, communication and automation technologies (ICATs) is accelerating the dramatic progresses of materials in life-cycle management. It is understood that it is essential for the composition–processing–structure–property–performance (CPSPP) relationship [4,6,7] to be comprehensively investigated in order to design advanced materials, and thus to manufacture their products. Taking the advantages of those digital technologies within ICATs, including high-performance high-throughput computations [8,9], big data and data mining [9,10], artificial intelligence [11,12,13], cloud and cloud manufacturing, and so on, the paradigm of discovery and development of advanced materials have been boosted from design to operation/manufacturing. On one hand, the multiscale and multidimension simulations have become important methods for the whole life-cycle management [4,14,15]. Advanced structural metal materials have been developed or manufactured crossing multiscales, from electronics to phases [16,17], atoms to autos [18], CALPHAD to flight [19]. On the other hand, the emerging data analytics is being employed for data preprocessing, analysis, decision making and visualization, resulting in the efficient creation/discovery of new materials and their robust and smart manufacturing [9,10,20]. It has been reported that the smart manufacturing multiplex is capable of executing multiple process chains and thermodynamic pathways to control the geometric, morphological and microstructural integrity of custom components [21]. Through efficiently and effectively utilizing the data and information across the CPSPP and the manufacturing procedures, the gaps between researchers and industries could be overlapped, resulting in a continuously enhanced active progress via the so-called digital twin procedure [15,22,23]. A suitable interface to interact with real data is available in cyber-physics systems, in which real data can be used as verification input for the simulation models and continuously lead to the further improvement [23]. In order to support the best optimized design or strategy based on life-cycle data, it is essential to simultaneously and comprehensively investigate the interrelating mechanisms between structure–form and structure–performance besides the material–property relationship.

It is worth mentioning that the digital twin, referring to the digital representation of physical objects, provides a powerful way to monitor and control assets and processes, which will be different from previous computer-aided design models [24]. Its main characteristics are important to address the aforementioned structure–form and structure–performance interaction mechanisms, which include (i) integrating various types of data of physical objects; (ii) existing in the entire life cycle of physical objects, coevolving with them and continuously accumulating relevant knowledge; and (iii) describing and optimizing physical objects [10,15,25]. For instance, as one of the classical digital twin models, the emergence of building information modeling (BIM) technology [26,27,28,29] has accelerated the dramatic evolution of the construction industry, supporting decision making about a project during its life cycle with efficient time, reasonable cost and improved quality. Since the steel mega-sub controlled structural system (MSCSS) is a typical model for designing safe tall and supertall buildings with good resistance to horizontal forces acted by wind and earthquakes [30,31], it has been applied in the construction of the NEC Office Building (180 m) and TC Tower (103 m) in Japan, the China Bank Building (315 m) in Hong Kong, the Shanghai Stock Building (128 m), and so on. One of the engineering challenges in the design of MSCSSs is to guarantee their stiffness under extreme conditions, such as earthquake and wind loads, and to maintainthe safety and comfort [32]. The dynamic magnification factor has been addressed as one kind of key structure dynamic response index in designing MSCSSs to significantly improve the self-absorption of shock [33]. The optimal parameters of the damper and its optimal arrangement play an important role affecting the control effectiveness of MSCSSs [34,35,36]. Moreover, with the aid of a rubber bearing at the top of additional column, the mechanical behavior of the column can be improved [34]. In particular, the acceleration response of the megaframe structure and the substructure can be further reduced [34].

In the present work, the steel MSCSS is selected as the case study to reveal the structure–property–performance relationship of a designed steel MSCSS under reported severe earthquake waves. Motivated by the digital twin strategy, the seismic responses of the designed MSCSSs are investigated theoretically and experimentally. In particular, comparing them with the traditional megaframe structure (MFS) used in the construction of Tokyo City Hall, the digital twin assistant active design and optimization of basic structural arrangements of MSCSSs are comprehensively investigated by integrating finite element simulations and experimental validations in order to support an optimized candidate ingenious configuration. In line with the concepts of data mining and life-cycle management, three natural severe earthquake waves, such as El Centro in 1940 (USA), TAFT in 1952 (USA) and Tangshan in 1976 (China), are set as the essential conditions to study the seismic response problems of various arrangements of MSCSSs, thus providing the best optimized configuration. This work supports a digital twin assistant active design and optimization strategy to further improve the control effectiveness of the system and to enhance the mechanical performance of the optimized configuration of the steel MSCSS.

## 2. Methodologies

### 2.1. Digital Twin Assistant Active Design and Optimization of MSCSSs

Motivated by the digital thread of lifetime management [15,37], the strategy of digital twin assistant active design and optimization of MSCSSs is proposed, which will comparing the theoretical and experimental principles/properties simultaneously. Particularly, the properties of designed MSCSSs will be further validated by the experimental tests, presenting a twinning feature between the cyber and the physical system. It is worth mentioning that the scaling rules together with the material properties, geometrical features and dynamic properties are comprehensively considered in the whole process, which are listed in Table 1 and reported in our previous work [35].

### 2.2. Finite Element Modeling

In line with the national standard “GB/T 50011-2010 Code for Seismic Design of Buildings”, there are at least three earthquake records required to be verified in designing earthquake-resistant structures. Here, the SAP2000 software [38] is utilized to investigate the response of MSCSS to three natural severe earthquake waves, including El Centro in 1940 (Imperial Valley, CA, USA), TAFT in 1952 (Kern County, CA, USA) and Tangshan in 1976 (Tangshan City, China) in terms of the modal analysis, the acceleration and the velocity of every floor and each subframe. In line with our previous simulations [35,39] and taking the advanced features of the SAP2000, complex models can be generated and meshed with powerful built-in templates. These aforementioned three seismic loads are constructed in the seismic load pattern form and applied to concentrate forces and moments at the bottom structures and along the frame elements. The constitutive equation based on the bilinear model [40] was utilized to calculate the repose of these designed Q235 steel MSCSS to the seismic load. With the aid of the integrated advanced analytical techniques, the deformation analysis is completed based on a stiffness of linear cases. While the viscous damper is utilized, the Maxwell model considering the performance of the viscous damper by frequency is used in the finite element calculations [39]. Correspondingly, the damping force can be expressed as [39]
(1)$$F_{d}\left(t\right)+\lambda \dot{F_{d}}\left(t\right)=C\left(\omega \right)\dot{u}\left(t\right)$$
where $F_{d}\left(t\right)$ is the damping force of the damper; $\dot{u}$ denotes the relative movement velocity of the thrust rod end of the damper with respect to the cylinder shell; $C\left(\omega \right)$ is the damping coefficient of the viscous damper, and its expression is as Equation (2); the linear damping coefficient at zero frequency is *C*_0_.
(2)$$C\left(\omega \right)=\frac{C_{0}}{1+\lambda^{2}\omega^{2}}$$
where λ can be calculated by Equation (3):(3)$$\lambda =\frac{C_{0}}{k_{d}}$$

When the performance of the damper mainly depends on the frequency, its support stiffness can be calculated according to Equation (4):(4)$$k_{d}=\left(6\pi /T_{1}\right)C_{V}$$
where $k_{d}$ is the support stiffness of the damper in the force direction; $T_{1}$ is the basic natural vibration period of the structures; $C_{V}$ is the linear damping coefficient. Those essential parameters of the damping and stiffness utilized in the finite element modeling are summarized in Table 2. It is worth mentioning that the best setting of the damping coefficient and the stiffness are 0.22 kN s/mm and 1.492 kN/mm to yield the best performance after systematic estimation and optimization, the results of which are discussed in the following.

### 2.3. Hilbert–Huang Transform (HHT)

Since Hilbert-Huang transform could efficiently reveal the nonstationary signals and precisely express the local time–frequency characteristics of the signal [38,39], it is used to construct the artificial waves based those aforementioned natural ones. The goal of the EMD is to decompose the signal into several inherent modal functions (IMF), thus meeting the requirements of performing a better Hilbert transformation. This procedure consists of the empirical model decomposition (EMD) and the Hilbert spectrum analysis [38,39], including (i) addressing the maximum and minimum envelopes through performing three spline interpolations of the corresponding maximum and minimum values of the original data *x*(*t*); (ii) connecting the averaged the upper and lower envelopes to yield the mean line *m*(*t*); and (iii) acquiring a new sequence *h*(*t*) without low frequencies through the following relationship:*H*(*t*) = *x*(*t*) − *m*(*t*) (5)

It is understood that the IMF yielded from EMD decomposition has a clear physical significance and can accurately calculate the instantaneous frequency of the signal [39].

Moreover, the Hilbert spectrum analysis is applied to analyze the instantaneous frequency and the amplitude of the IMF referring to the time and frequency through the so-called Hilbert transform expressed as [38,39].
(6)$$y\left(t\right)=\frac{1}{\pi }\int \frac{x\left(\tau \right)}{t-\tau }d\tau$$
(7)$$Ζ\left(t\right)=x\left(t\right)+iy\left(t\right)$$
(8)$$z\left(t\right)=\alpha \left(t\right)e^{i\theta \left(t\right)}$$
(9)$$\alpha \left(t\right)=\sqrt{x^{2}\left(t\right)+y^{2}\left(t\right)}$$
(10)$$\theta \left(t\right)=arctan\frac{y\left(t\right)}{x\left(t\right)}$$
(11)$$\omega \left(t\right)=\frac{d\theta \left(t\right)}{dt}$$
(12)$$H\left(t\right)=\overset{n}{\underset{j=1}{\sum }}\alpha_{j}\left(t\right)exp [i\int w_{j}\left(t\right)dt]$$
where *c_j_*(*t*) is the *j*-th order modal function. The *z*(*t*) and *a_j_*(*t*) denote the analytical signal of *x*(*t*) and its amplitude of the *c_j_*(*t*), respectively. Therefore, the time–frequency characteristics of the signal *x*(*t*) can be obtained, presenting all amplitudes in time and frequency space [38,39]. The so-called acceleration response *β*, along the mega-sub structural floor number to various severe earthquake waves, was utilized to further characterize the damage probability matrix, which can be conveniently expressed in terms of a probability density function [38].

## 3. Results and Discussions

### 3.1. Optimized Configurations of MSCSS Together with Numerical Analysis

The conventional MSCSS is consisted of two major components. While the megaframe is the dominating structural frame of one building, the other series of substructures plays a key role in commercial and/or residential applications [31,32,38]. It is worth mentioning that the structural control principles and mechanisms should be treated seriously during the design of conventional MSCSSs [32]. In the present work, the main optimized parameters are the sub- and megastructure mass ratio and the substructure stiffness ratio. It is understood that those parts or elements of the mega-substructure would convert the traditional MFS into a huge, self-controlled, passive MSCSS that can yield the energy dissipation responding to those from the natural forces [31,32], as shown in Figure 1a,b. In line with the design principle in term of the sub- and megastructure mass ratio, the weight of the designed traditional MFS labeled as MFS1 is 74.8 kg if utilizing the Q235 steel wires and tubes, as shown in Figure 1a,c. As for the MFS2 in Figure 1c, besides an extra weight of about 6 kg yielded by the viscous dampers and the connectors on the top of the frame, there is no change comparing with MFS1. Moreover, three MSCSSs are designed to be investigated in the present work, as shown in Figure 1c. In particular, the MSCSS001 configuration consists of four megafloors, with a uniform number and arrangement of a “modulated substructure” (seven stories) at the top of the megafloor and three “attached substructures” at the megafloors [31]. Similarly, the MSCSS010 and MSCSS100 configurations are constructed by inserting a “modulated substructure” at different substructures, in which the subframes are capable of absorbing the energy during an earthquake and to transfer the vertical load to the megaframes [31]. It is worth mentioning that the control system is cost-effective since there is no additional mass required to perform its efficiency [31].

### 3.2. The Benchmarks of the Ground Motion Waves Together with Their HHTs

Figure 2 presents the comparations of the acceleration response between the artificial waves and their reference natural ones, including the El Centro wave, the TAFT wave, and the Tangshan wave. In general, it can be seen that the artificial waves generated via the HHT match these natural ones well and are controlled precisely, which is essential to estimate the response of MSCSSs to the ambient excitations, especially the seismic waves. Based on the record of the El Centro wave, it was maintained for 53.73 s. The maximum accelerations of the meridian direction, the latitude direction and the radial direction were 341.70 mm s^−2^, 210.10 cm s^−2^ and 206.30 cm s^−2^, respectively, as shown in Figure 2a. The maximum acceleration was more than 3000 mm s^−2^ at the initial time, followed by several peaks around 2000 mm s^−2^ within 20 s. Similarly, the TAFT wave was kept for 54.38 s. Its maximum accelerations of the southeast direction, the northeast direction and the radial direction were 175.95 mm s^−2^, 152.70 cm s^−2^ and 102.85 cm s^−2^, respectively, as shown in Figure 2b. The Tangshan Wave was maintained for 23.19 s, with its maximum accelerations of the meridian direction, the latitude direction and the radial direction being 158.62 mm s^−2^, 150.39 cm s^−2^ and 79.04 cm s^−2^, respectively, as shown in Figure 2c. It is noted that there were a great number of acceleration peaks above 1000 mm s^−2^ within the first 5 s of Tangshan Wave, which yielded a severe disaster due to the short response time for humans. Therefore, the frequencies and amplitudes of natural waves are extremely important in precisely reproducing the artificial ones via the HHT, both of which have been utilized in training the deep learning model to solve pixel-to-pixel tasks [41,42].

### 3.3. Structure Response Characteristics and Its Optimizations

Figure 3 displays the acceleration response of the investigated five configurations along the mega-sub structural floor number to various severe earthquake waves, the dominate/key frequencies of which are listed in Table 3. While the first six order modes are analyzed theoretically, the first three ones, namely translation, translation and torsion, are selected to display the stiffness distributions of the designed configurations referring to the classical MFS. It is understood that the ratio of the torsional period over the translational period should be less than 0.9 to fit the requirements of design specification [39]. In fact, there are various principles and innovative strategies utilized to reveal the seismic isolation and energy dissipation and to design the MSCSS, including the SSSA method, the nonrepetitive method, the energy-based stochastic approach integrated with novel equal-energy non-Gaussian SLT, and so on [39]. Through characterizing those fundamental properties in terms of the reduction in absolute acceleration, peak interstory drift and residual drift, the advantages of the standards and the advanced viscous damper placement methods can be revealed comprehensively. As presented in Table 3, it is noted that the first six order frequencies of MFS2 configurations are lower than those of MFS1 ones, which is caused by the extra mass on the top acting as a damper and present the reduced displacement in the 3rd-order model in Figure 3. Correspondingly, referring to the benchmark configuration MFS1, the recent designed MSCSS0001 configuration presented the best performance since all those six order frequencies were decreased dramatically.

As shown in Figure 4, comparing to the configurations of MFS1 and MFS2, the dramatical reductions in the vibration responses of MSCSS0001, MSCSS0010 and MSCSS0100 configurations are presented in terms of acceleration velocity factor (β). Since MFS1 is selected as the benchmark, the corresponding results of optimized configurations under the El Centro, TAFT and Tangshan waves are listed in Table 4, Table 5, Table 6, Table 7, Table 8 and Table 9 in detail. It can be found that the MSCSS configurations are capable of effectively optimizing the vibration responses with significantly decreased acceleration, which is also much better than MFS2 with extra weight. Moreover, if the horizontal connections between the sub- and the megastructures are broken, the displacement of the megastructure would be smaller than that of the substructure. The larger displacements or the obvious response of the substructures should be caused by the extra weight of the damper on the top floor, which improves the mass ratio between the damping and the MSCSS structure and drives the application of the TMD theory. Correspondingly, the response of the megastructure will be dramatically reduced by the effects of the substructure on the energy dissipation and the response frequency, which is further aided by the damping. It is worth mentioning that the formation of abrupt amplified β of the top floors should be attributed to the sheath effect. Furthermore, the displacement of the substructure is one kind of energy dissipation. Its larger displacement will result in a greater amount of energy dissipation and better performance of the designed configuration. For instance, the top floor of the substructure of MSCSS0001 displays the smallest value of β prefactor responding to all these three severe earthquake waves, indicating that this the best-designed MSCSS configuration to be further validated experimentally.

Based on these theoretical results listed in Table 4, Table 5, Table 6, Table 7, Table 8 and Table 9, it is understood that the response of a given configuration to various severe earthquake waves will present different behaviors/performances. The response control effect of MSCSS is much better than those of MFS ones, which matches well with previous reports presenting the excellent performance of optimized MSCSS [33,35]. However, the response control effect of the MSCSS will be decreased if the positions of those substructures are designed at the lower floors. On the contrary, the best performance of MSCSS can be obtained only if those substructures are at the bottom. The damper in the megastructure will furtherly improve the response control effect of MSCSS, which is recommended to be investigated in the future work.

### 3.4. Experimental Validation

With the guidance of the national standard named “GB/T342-1997 Dimension Shape Mass and Tolerance for Cold-Drawn Round Square and Hexagonal steel Wires”, the aforementioned best configuration, named MSCSS0001, was manufactured by the Q235 steel, as shown in Figure 5. It consists of 36 floors, which can also be divided into 4 substructures with 7 interfloor spacings and 1 megastructure. Since the ratio between the natural and the designed model is 1:100, the interfloor space of the substructure is 100 mm, while it is 80 mm for the megastructure and the standard ones, yielding a total MSCSS0001 frame length of 2.96 m, as displayed in Figure 5a. As can be seen in Figure 5b–d, the front view, the side view and the top view of the schematic diagram together with the arrangements of sensors are displayed, respectively. Correspondingly, the comparations of the theoretical and experimental frequencies together with their error bar are listed in Table 10, presenting a good agreement. For the benchmark tests of the MFS structure, it is noted that the error bars of four order models are within 2%, while the error bars of the 3rd-and 6th-order models are 22.98% and −12.04%, respectively. On the contrary, the error bars of the 1st-, 2nd- and 6th-order models of the MSCSS are within 10% when comparing them to the experimental ones. The error bar of the 3rd-order model is as high as 45.12%, which is caused by the manually applied torsion forces completed along the X and Y directions separately. The coupling effects of those errors along the X and Y directions will be amplified higher than each individual case. Moreover, the residual stresses of the frame along these two directions will be different and asymmetric, which could cause the extra error for the torsion analysis, while the nonlinear numerical calculations are utilized in the analysis of the dynamic response of MSCSSs, the ideal states without considering the contributions of joints among the sub-subframes, the sub-megaframes and the mega-megaframes, which may cause the damage/collapse of the real structure. Since the nonlinear analysis is dominated by the plastic hinge, its formation will characterize the damage of the natural MSCSS during the experimental measurements, which will render comparisons among these designed MSCSSs via the same basic matrix impossible. Correspondingly, it is strongly recommended that the linear analysis should be completed theoretically and experimentally to compare their different performance precisely and consistently. Although the natural model did not suffer damage, it is essential to highlight that the computer simulation of the structural behavior could significantly underestimate the response values of the experimentally tested model. This is very dangerous, if everything is put in the context of real constructions.

Figure 6 and Figure 7 present the response of the typical structural unit of MSCSS0001 to the El Centro wave in views of velocity and acceleration, respectively. It can be seen that our theoretically calculated velocity and acceleration of the top megastructure and substructure frames agree well with the experimental ones. In particular, as shown in Figure 6a,b, the velocities of the first top megastructure and substructure frames are 0.0358 m s^−1^ and 0.0359 m s^−1^, revealing the control efficiency of 48.930% and 47.744%. As for the second and third top megastructure and substructure frames, as shown in Figure 6c–f, the error bar of the higher layers will be larger than that of the lower layer (i.e., the first one) since there are some differences referring to those of the experimental ones. Similarly, the theoretical acceleration response of the typical structural unit of MSCSS0001 to the El Centro wave agrees well with the experimental ones, although there is a little bit of difference on the peak values within the first 7 s, as displayed in Figure 7. Therefore, the shapes and frequencies of the theoretical and experimental results are similar, indicating that the accurate and reliable responses of our designed MSCSS configurations have been validated in the view of velocity and acceleration.

## 4. Conclusions

In the present work, the structure–property–performance relationship of the designed steel MSCSS under the reported earthquake waves has been investigated through integrating the finite element simulations and the experimental validations. Through characterizing those fundamental properties in terms of the reduction of absolute acceleration, peak interstory drift and residual drift, the advantages of the standards and the advanced viscous damper placement methods can be revealed comprehensively. While the first six order modes of MSCSS were analyzed theoretically, the first three ones, namely translation, translation and torsion, were selected to display the stiffness distributions of the designed configurations referring to the classical MFS. It is understood that the ratio of the torsional period over the translational period should be less than 0.9 to fit the requirements of design specification. It is noted that the first six order frequencies of MFS2 configurations were lower than those of MFS1 ones, which was caused by the extra mass on the top acting as a damper and presented the reduced displacement in the 3rd-order model. Correspondingly, referring to the benchmark configuration MFS1, the recent designed MSCSS0001 configuration presented the best performance since all six of those order frequencies decreased dramatically. The response control effect of the MSCSS was much better than those of the MFS ones, which matches well with previous reports presenting the excellent performance of optimized MSCSS [33,35]. However, the response control effect of the MSCSS would decrease if the positions of those substructures were designed at the lower floors. On the contrary, the best performance of MSCSS can be obtained only if those substructures are at the bottom. The MSCSS configurations are capable of effectively optimizing the vibration responses with significantly decreased acceleration, which is also much better than MFS2 with extra weight. Moreover, the displacement of the megastructure would be smaller than that of the substructure if the horizontal connections between the sub- and the megastructures were broken. It is highlighted that the displacement of the substructure is one kind of energy dissipation, yielding a greater amount of energy dissipation and better performance of the designed configuration. Therefore, the present proposed strategy of digital twin assistant active design and optimization of MSCSS supports a decision-making approach, accelerating its development with efficient time, reasonable cost and improved quality.

## Figures and Tables

**Figure 1 materials-15-06382-f001:**
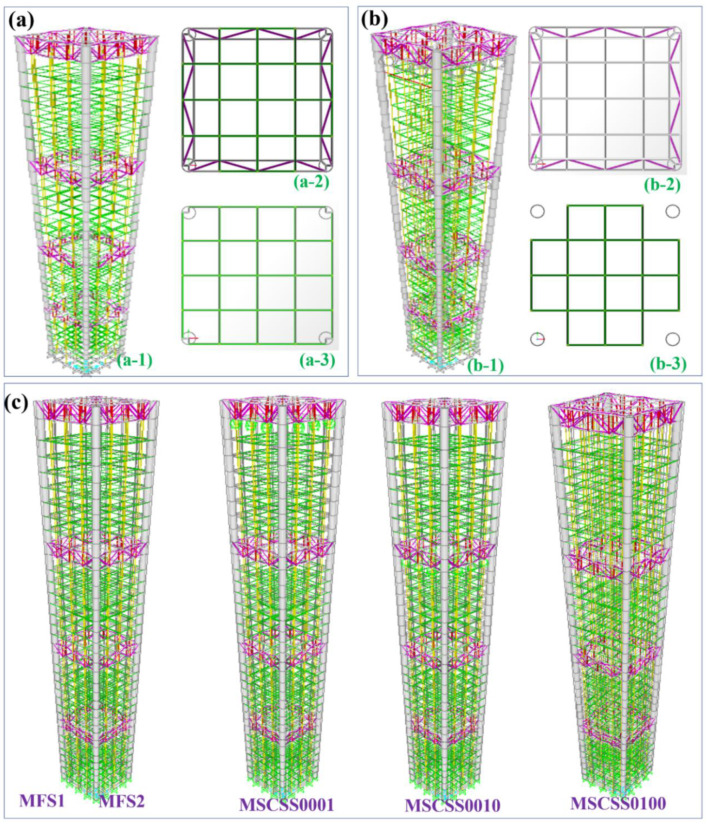
The geometry of designed steel MSCSS configurations. (**a**) The traditional megaframe structure (MFS) together with top views of the megaframe and the subframe; (**b**) the MSCSS together with top views of the megaframe and the optimized subframe; (**c**) the designed MSCSS configurations referring to the classical MFS ones.

**Figure 2 materials-15-06382-f002:**
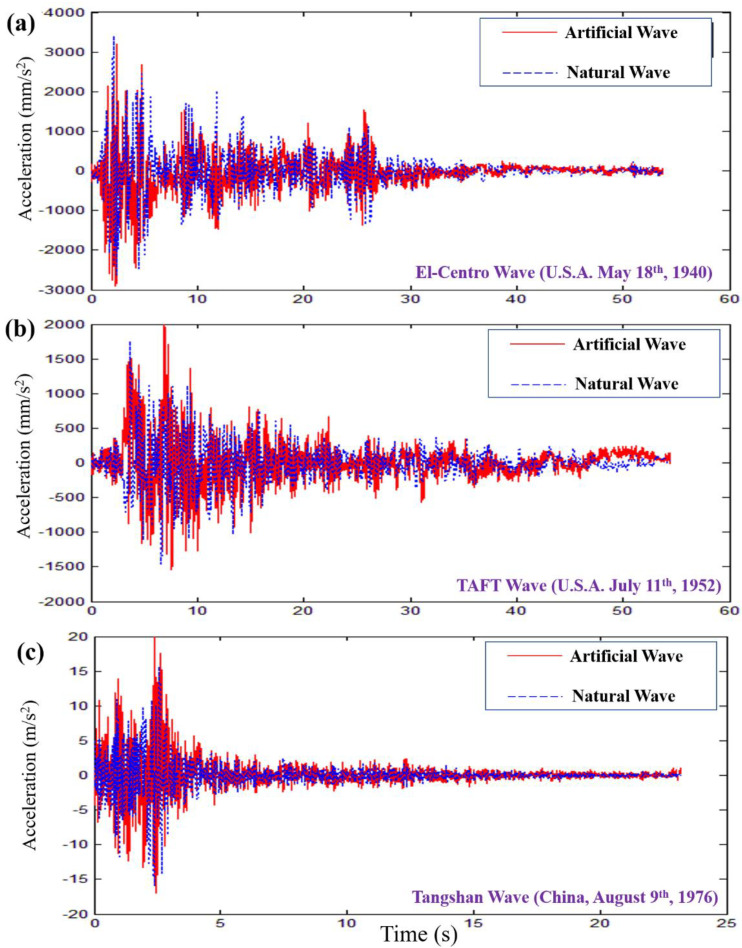
The comparations of the acceleration response between the artificial waves and their reference natural ones. (**a**) El Centro wave; (**b**) TAFT wave; (**c**) Tangshan Wave.

**Figure 3 materials-15-06382-f003:**
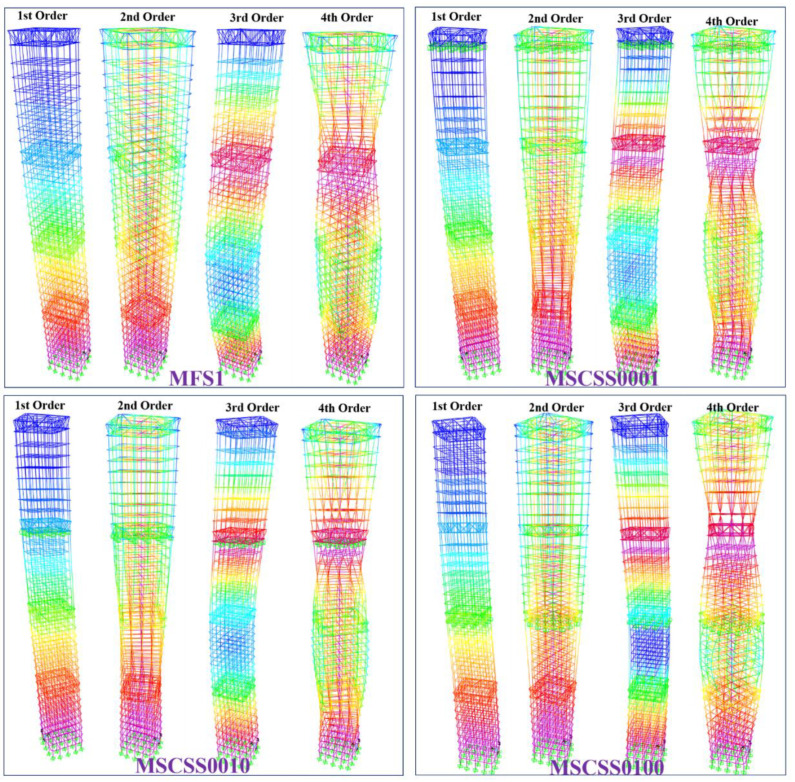
The modal analysis of the designed MSCSS configurations together with the conventional MFS one.

**Figure 4 materials-15-06382-f004:**
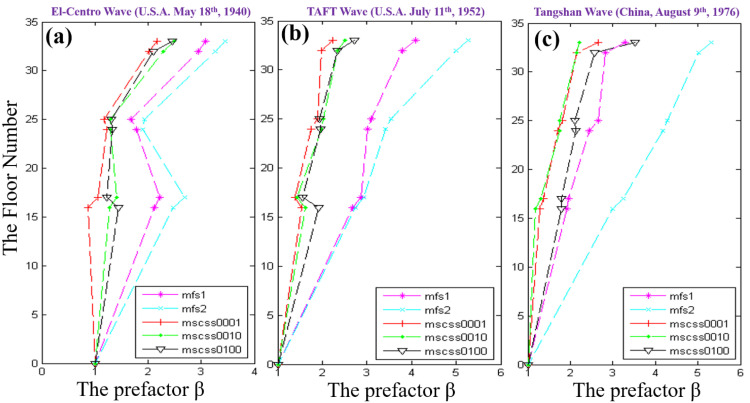
The acceleration response along the mega-sub structural floor number to various severe earthquake waves: (**a**) the El Centro; (**b**) the TAFT; (**c**) the Tangshan.

**Figure 5 materials-15-06382-f005:**
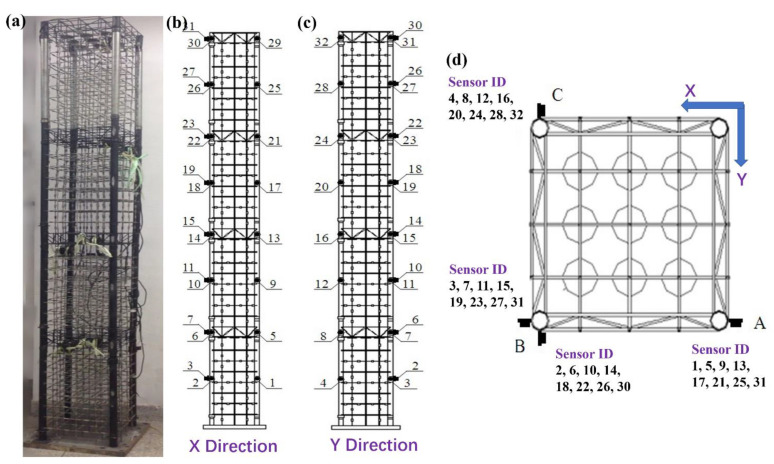
The investigated best candidate MSCSS0001 configuration together with the sensor arrangement. (**a**) The fabricated steel frame of the investigated best candidate configuration MSCSS0001 in the ratio of 1:100; (**b**–**d**) The front view, the side view and the top view of the schematic diagram together with the arrangements of sensors, respectively. These numbers from 1 to 32 highlight the labeled tags/identifiers of utilized sensors in different position.

**Figure 6 materials-15-06382-f006:**
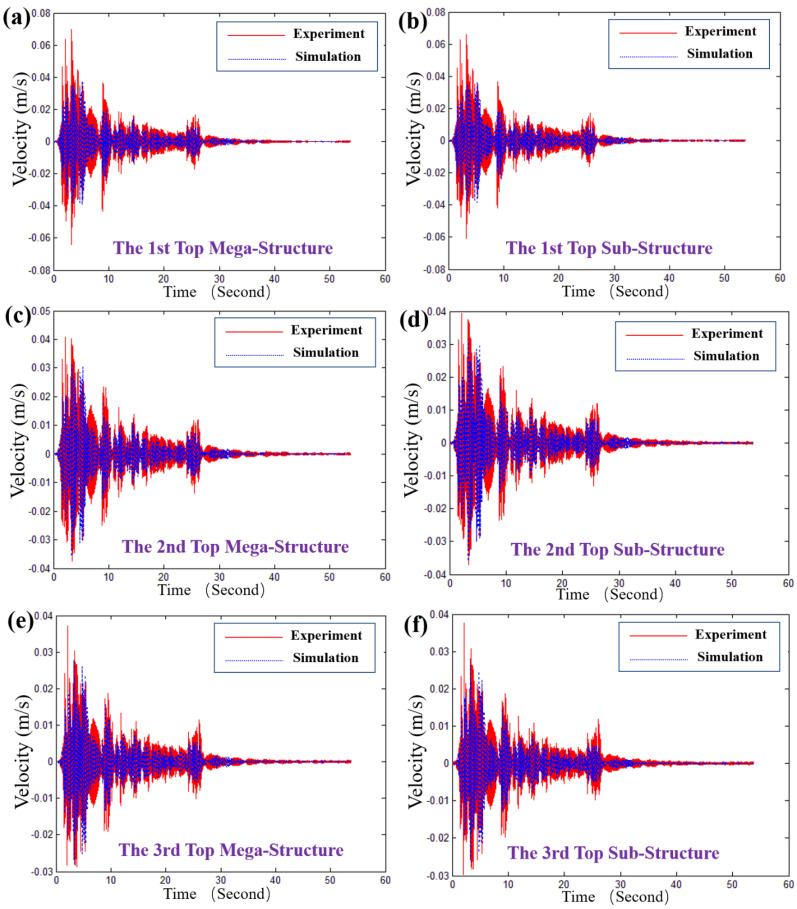
The response of the typical structural unit of MSCSS0001 to the El Centro wave: (**a**) the top of the first megaframe; (**b**) the top of the first substructure; (**c**) the top of the second megaframe; (**d**) the top of the second substructure; (**e**) the top of the third megaframe; (**f**) the top of the third substructure.

**Figure 7 materials-15-06382-f007:**
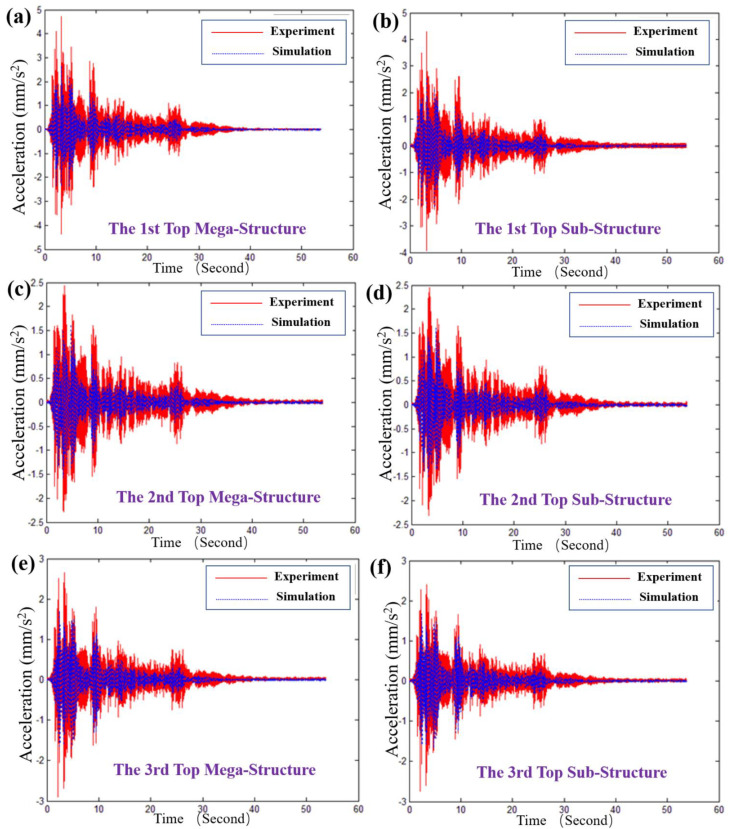
The acceleration response of the typical structural unit of MSCSS0001 to the El Centro wave: (**a**) the top of the first megaframe; (**b**) the top of the first substructure; (**c**) the top of the second megaframe; (**d**) the top of the second substructure; (**e**) the top of the third megaframe; (**f**) the top of the third substructure.

**Table 1 materials-15-06382-t001:** Similarity relation between the design model and the real MSCSS [35].

	Quantities	Symbol	Similarity	Ratio (Real Structure/Virtual Model)
**Materials** **Properties**	Stress	$S_{\sigma }$	$S_{\sigma }=S_{E}$	1.0
	Strain	$S_{\varepsilon }$	$S_{E}=1.0$	1.0
	Elastic Modulus	$S_{E}$	$S_{E}$	1.0
	Poisson Ratio	$S_{\mu }$	$S_{\mu }=1.0$	1.0
	Density	$S_{\rho }$	$S_{\rho }=S_{E}/S_{L}$	100
**Geometrical** **Features**	Length	$S_{L}$	$S_{L}$	0.01
	Area	$S_{A}$	$S_{A}=S_{L}^{2}$	0.0001
	Linear Displacement	$S_{X}$	$S_{X}=S_{L}$	0.01
	Angular Displacement	$S_{\theta }$	$S_{\theta }=1.0$	1.0
**Dynamic** **Properties**	Mass	$S_{m}$	$S_{m}=S_{\rho }S_{L}^{3}=S_{E}S_{L}^{2}$	0.0001
	Stiffness	$S_{k}$	$S_{k}=S_{E}S_{L}$	0.01
	Intrinsic Cycle	$S_{T}$	$S_{T}={\left(S_{m}/S_{k}\right)}^{1/2}$	0.1
	Frequency	$S_{f}$	$S_{f}=1/S_{T}=S_{L}^{-0.5}$	10
	Damping	$S_{c}$	$S_{c}=S_{m}/S_{T}=S_{E}S_{L}^{1.5}$	0.001
	Velocity	$S_{v}$	$S_{v}=S_{X}/S_{t}=S_{L}^{0.5}$	0.1
	Acceleration	$S_{a}$	$S_{a}=S_{X}/S_{T}^{2}=1.0$	1.0
	Gravity	$S_{g}$	$S_{g}=1.0$	1.0

**Table 2 materials-15-06382-t002:** Those essential parameters of the damping and stiffness utilized in the finite element modeling.

Damping Coefficient (kN s/mm)	**0.22**	0.33	0.55	1.1	5.5	11
Stiffness(kN/mm)	**1.492**	2.24	3.73	7.47	37.3	74.7

**Table 3 materials-15-06382-t003:** The theoretical frequencies (Hz) of investigated configurations.

Configuration	1st-Order Mode	2nd-Order Mode	3rd-Order Mode	4th-Order Mode	5th-Order Mode	6th-Order Mode
**MFS1**	6.9454	6.9454	17.4612	21.5471	21.5471	36.6435
**MFS2**	6.4599	6.4599	17.4600	20.3890	20.3890	34.4588
**MSCSS0001**	5.6767	5.6767	17.4703	18.2615	18.2615	22.6706
**MSCSS0010**	6.4510	6.4510	18.3690	21.3575	21.3578	29.3068
**MSCSS0100**	6.4616	6.4616	13.4373	21.4225	21.4225	31.2013

**Table 4 materials-15-06382-t004:** The velocity response at a critical point of the investigated configurations under the El Centro wave (m s^−1^).

Configuration	1st Megastructure Top Floor	1st Substructure Top Floor	2nd Megastructure Top Floor	2nd Substructure Top Floor	3rd Megastructure Top Floor	3rd Substructure Top Floor
**MFS1**	0.0701	0.0687	0.0566	0.0550	0.0448	0.0424
**MSCSS0001**	0.0358	0.0359	0.0308	0.0306	0.0270	0.0257
**MSCSS0010**	0.0378	0.0369	0.0317	0.0315	0.0278	0.0264
**MSCSS0100**	0.0598	0.0582	0.0485	0.0523	0.0384	0.0370

**Table 5 materials-15-06382-t005:** The velocity response at a critical point of the investigated configurations under the Tangshan wave (m s^−1^).

Configuration	1st Megastructure Top Floor	1st Substructure Top Floor	2nd Megastructure Top Floor	2nd Substructure Top Floor	3rd Megastructure Top Floor	3rd Substructure Top Floor
**MFS1**	0.3135	0.3069	0.2479	0.2381	0.1612	0.1494
**MSCSS0001**	0.2169	0.2174	0.1877	0.1821	0.1329	0.1246
**MSCSS0010**	0.2248	0.2204	0.1921	0.1862	0.1341	0.1256
**MSCSS0100**	0.1582	0.1539	0.1288	0.1361	0.0963	0.0903

**Table 6 materials-15-06382-t006:** The velocity response at a critical point of the investigated configurations under the TAFT wave (m s^−1^).

Configuration	1st Megastructure Top Floor	1st Substructure Top Floor	2nd Megastructure Top Floor	2nd Substructure Top Floor	3rd Megastructure Top Floor	3rd Substructure Top Floor
**MFS1**	0.0356	0.0349	0.0291	0.0280	0.0186	0.0174
**MSCSS0001**	0.0287	0.0287	0.0244	0.0236	0.0166	0.0155
**MSCSS0010**	0.0296	0.0291	0.0248	0.0240	0.0169	0.0157
**MSCSS0100**	0.0348	0.0343	0.0298	0.0302	0.0201	0.0188

**Table 7 materials-15-06382-t007:** The acceleration response at a critical point of the investigated configurations under the El Centro wave (m s^−2^).

Configuration	1st Megastructure Top Floor	1st Substructure Top Floor	2nd Megastructure Top Floor	2nd Substructure Top Floor	3rd Megastructure Top Floor	3rd Substructure Top Floor
**MFS1**	4.77958	4.56826	3.26028	3.25423	3.0092	2.92027
**MSCSS0001**	2.54298	2.55961	1.61258	1.58138	1.66917	1.64302
**MSCSS0010**	2.66669	2.58399	1.62786	1.5999	1.70185	1.67041
**MSCSS0100**	3.60737	3.35749	1.92975	2.22256	2.41596	2.41865

**Table 8 materials-15-06382-t008:** The acceleration response at a critical point of the investigated configurations under the Tangshan wave (m s^−2^).

Configuration	1st Megastructure Top Floor	1st Substructure Top Floor	2nd Megastructure Top Floor	2nd Substructure Top Floor	3rd Megastructure Top Floor	3rd Substructure Top Floor
**MFS1**	13.28673	12.98759	10.5392	10.23442	12.5419	12.23583
**MSCSS0001**	11.54402	11.57088	8.31653	7.80441	11.27876	11.02887
**MSCSS0010**	11.91685	11.62903	8.63159	8.15865	11.95438	11.67911
**MSCSS0100**	9.28152	9.02504	6.52034	6.75927	6.43695	6.55522

**Table 9 materials-15-06382-t009:** The acceleration response at a critical point of the investigated configurations under the TAFT wave (m s^−2^).

Configuration	1st Megastructure Top Floor	1st Substructure Top Floor	2nd Megastructure Top Floor	2nd Substructure Top Floor	3rd Megastructure Top Floor	3rd Substructure Top Floor
**MFS1**	1.53296	1.49571	1.43937	1.40308	1.00678	0.93036
**MSCSS0001**	1.33797	1.34093	1.0238	0.96878	0.85073	0.81348
**MSCSS0010**	1.41850	1.38444	1.00713	0.96938	0.89715	0.85761
**MSCSS0100**	1.31576	1.29558	1.09562	1.16114	0.72263	0.67056

**Table 10 materials-15-06382-t010:** The comparations of the theoretical and experimental frequencies (Hz) together with their error bar.

Frequency	MFS1			MSCSS0001		
	SAP2000	Experiment	Error Bar	SAP2000	Experiment	Error Bar
**1st-order model**	6.945409	6.783	2.34%	5.676658	5.130	9.63%
**2nd-order model**	6.945409	6.883	0.90%	5.676658	5.313	6.41%
**3rd-order model**	17.46115	13.448	22.98%	17.4703	9.587	45.12%
**4th-order model**	21.54708	21.687	−0.65%	18.2615	13.716	24.89%
**5th-order model**	21.54708	21.810	−1.22%	18.2615	14.166	22.43%
**6th-order model**	36.64346	41.056	−12.04%	22.6706	21.811	3.79%

## Data Availability

Correspondence and requests for data and materials should be addressed to W.Y.W. (wywang@nwpu.edu.cn) and X.A.Z. (jiaoping@nwpu.edu.cn).

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
