# Peer review of "Digital Twin Assistant Active Design and Optimization of Steel Mega-Sub Controlled Structural System under Severe Earthquake Waves"

_materials, 2022, doi:10.3390/ma15186382_

Round 1
Reviewer 1 Report
This paper deals with the dynamic performance of a steel structure. The introduction is very interesting and comprehensive. The work is well written, and the conducted experiment for which behavioral assessment was carried out is particularly commendable. However, before deciding to accept the paper, I ask the authors to address the following issues:
Minor issues:
Please change the key words and remove the abbreviations from the key words. Keywords should be simple and reflect the work. I suggest choosing keywords from the title and abstract.
I advise the authors to find a native English speaker to proofread the manuscript.
It is necessary to provide a reference for the computer program, so that it is clear in which version and with what capabilities the computer model was created.
Please explain (best using tables) what values for damping and stiffness were included to the computer model.
Instead of the word speed, please use the word velocity.
Mega-structure sounds odd. It is better to use the term super-structure and sub-structure. However, sub-structure mainly refers to the foundation and or soil-foundation system, which is not described here. Maybe I misinterpreted something. Please explain the choice of the term mega-sub structure.
The conclusion is too general and needs to be rewritten in its entirety.
The title does not reflect the work presented in the manuscript.
Major issues:
There is no detailed information regarding the structures observed. How the structure is modeled. What finite elements where used (and how)? How were the floor structure, foundation, joints, etc. modeled? What analysis type and solver were used? What kind of load on the structure is modeled (and according to which code/norm)? What kind of mesh was employed?
Please explain why exactly the three mentioned records of earthquakes, i.e. input excitations, were used. It is necessary to describe them in detail in the context of their properties (i.e. predominant period, frequency content, Arias intensity etc.) and show the spectrum of the acceleration response but also the Fourier spectrum for the selected records. I suggest using a program like SeismoSpect or SeismoSelect or similar to determine the properties of the records used.
Many international standards for the design of earthquake-resistant structures require the verification of behavior under the action of at least seven earthquake records. Please argue why only three records were used.
The calculation results (described in tables and graphically) should be interpreted with regard to the standard limit values.
When it comes to small-scale structural models, it is necessary to define/describe scaling rules. Without this, it is not possible to accept the submitted manuscript.
A detailed description of the geometry, mass and materials for the small-scale model is missing. It is necessary to describe in details the sensors that are placed on the physical small-scale model.
In the manuscript, it should be clearly written whether the work is done in linear-elastic or non-linear numerical calculations. Likewise, it should be clearly written if the physical model of the structure has suffered damage.
The computer simulation of the structural behavior significantly underestimates the response values of the experimentally tested model. This is very dangerous, if everything is put in the context of real constructions.
Specific issues:
Lines 99-100: Modal analysis has nothing to do with seismic loading. It is carried out in order to determine the dynamic properties of the structure.
Lines 231-232: Incorrect interpretation. It is necessary to look at the structure's response in more detail in the context of the shape of the structure's vibration and the frequency composition of the excitations.
Line 250-257: the comments are overstated and insignificant. Please change and reformulate.
Reviewer 2 Report
The study presents and discusses a digital-twin-assistant active design and optimization strategy regarding the MSCSS performance under earthquake waves. Undoubtedly, this subject is of great value, with many gaps regarding this subject in the literature. However, weaknesses and shortcomings are detected primarily in the methodological approach, where insufficient discussion and limited analysis are included.
Significant points in the article that need clarification or refinement:
- The authors should provide more compelling evidence of the paper's contribution to this field. Which are the comparative benefits/advantages of your finding? Although the importance of studying this subject is proven, it is not explicitly presented in the manuscript how this study and its results would be a novelty. It is a crucial point that needs to be reviewed in this work;
- Research Background: A more in-depth literature review is needed. It would be necessary for the authors to picture the problem better for the readers.
- Methodology and results: Although the authors have presented pertinent information about the impacts of severe natural earthquake waves, the article lacks an explanation of the concept and principles of the methods used in the study. In some excerpts, the methods used are not profoundly explained. How was the digital twin concept applied in this study in practice?
- The conclusion could be expanded, including the implications/recommendations of the study, limitations, and further studies, in addition to being supported by the results.
- Minor English proofreading is required.
Round 2
Reviewer 1 Report
The authors took into account a significant number of comments and suggestions. However, I still think that the numerical model is described in insufficient detail. The authors state that a nonlinear analysis was performed on a numerical model, but the description of the constitutive models is missing in the manuscript. Thus the repeatability of the numerical model is not ensured.
It is not enough to observe the response of such a complex structure through the time-histories of acceleration and/or velocity. I still believe that the behavior of the model should be looked at more deeply, especially through acceleration response spectra or the Fourier spectrum.
Reviewer 2 Report
The authors addressed all suggestions, and the paper significantly improved after this review. In my opinion, the article can be published in the presented form.
Author Response
Thanks again for the recommendation and those constructive suggestions/comments, which significantly improve the quality of the revised manuscript.
Round 3
Reviewer 1 Report
The authors improved the work by taking into account most of the suggested comments and corrections.